# Parkinson’s Disease and the Heart: Studying Cardiac Metabolism in the 6-Hydroxydopamine Model

**DOI:** 10.3390/ijms241512202

**Published:** 2023-07-30

**Authors:** Victor Silva da Fonsêca, Valeria de Cassia Goncalves, Mario Augusto Izidoro, Antônio-Carlos Guimarães de Almeida, Fernando Luiz Affonso Fonseca, Fulvio Alexandre Scorza, Josef Finsterer, Carla Alessandra Scorza

**Affiliations:** 1Disciplina de Neurociência, Departamento de Neurologia e Neurocirurgia, Universidade Federal de São Paulo (UNIFESP), São Paulo 04039-032, Brazil; fonseca_victor1@hotmail.com (V.S.d.F.); vaal.cassia@gmail.com (V.d.C.G.); scorza@unifesp.br (F.A.S.); 2Laboratório de Espectrometria de Massas-Associação Beneficente de Coleta de Sangue (COLSAN), São Paulo 04038-000, Brazil; mario.izidoro77@gmail.com; 3Laboratório de Neurociências Experimental e Computacional, Departamento de Engenharia de Biossistemas, Universidade Federal de São João del-Rei (UFSJ), São João del Rei 36301-160, Brazil; acga@ufsj.edu.br; 4Laboratório de Análises Clínicas da Faculdade de Medicina do ABC, Santo André 09060-650, Brazil; profferfonseca@gmail.com; 5Departamento de Ciências Farmacêuticas da Universidade Federal de Sao Paulo (UNIFESP), Diadema 09972-270, Brazil; 6Neurology & Neurophysiology Center, 1180 Vienna, Austria; fifigs1@yahoo.de

**Keywords:** Parkinson’s disease, cardiac metabolism, heart, 6-hydroxydopamine, metabolites

## Abstract

Parkinson’s-disease (PD) is an incurable, age-related neurodegenerative disease, and its global prevalence of disability and death has increased exponentially. Although motor symptoms are the characteristic manifestations of PD, the clinical spectrum also contains a wide variety of non-motor symptoms, which are the main cause of disability and determinants of the decrease in a patient’s quality of life. Noteworthy in this regard is the stress on the cardiac system that is often observed in the course of PD; however, its effects have not yet been adequately researched. Here, an untargeted metabolomics approach was used to assess changes in cardiac metabolism in the 6-hydroxydopamine model of PD. Beta-sitosterol, campesterol, cholesterol, monoacylglycerol, α-tocopherol, stearic acid, beta-glycerophosphoric acid, o-phosphoethanolamine, myo-inositol-1-phosphate, alanine, valine and allothreonine are the metabolites that significantly discriminate parkinsonian rats from sham counterparts. Upon analysis of the metabolic pathways with the aim of uncovering the main biological pathways involved in concentration patterns of cardiac metabolites, the biosynthesis of both phosphatidylethanolamine and phosphatidylcholine, the glucose-alanine cycle, glutathione metabolism and plasmalogen synthesis most adequately differentiated sham and parkinsonian rats. Our results reveal that both lipid and energy metabolism are particularly involved in changes in cardiac metabolism in PD. These results provide insight into cardiac metabolic signatures in PD and indicate potential targets for further investigation.

## 1. Introduction

The dramatic increase in the world’s elderly population brings with it an increased risk of developing neurodegenerative diseases [1]. Parkinson’s disease (PD) is a common age-related neurodegenerative disease identified as the fastest growing disease among neurological disorders in terms of disability and death [2]. The nigrostriatal dopaminergic system is particularly vulnerable in PD, leading to noticeable motor deficits. However, the disease is also known to cause a plethora of non-motor manifestations that impose a significant burden on patients, with an emphasis on cardiovascular abnormalities, which are prevalent in individuals with PD and associated with an increase in morbidity and mortality [3].

The precise mechanisms driving the onset and progression of PD remain elusive. Nonetheless, key pathological features, such as oxidative stress, mitochondrial dysfunction and neuroinflammation have been identified [4]. Redox disturbance manifests as an early biochemical change in PD. The pathogenesis of PD involves the impairment of mechanisms that counteract oxidative stress-induced damage by neutralizing reactive oxygen species, such as deficiencies in the glutathione system, leading to increased oxidative toxicity [5,6]. The neurodegenerative process is intricately connected to the disruption of homeostatic and defensive roles of glial cells as well as the acquisition of toxic glial properties [7]. Astrocytes and microglia play a pivotal role as initial responders to brain changes, releasing a wide range of bioactive molecules [8]. Various studies substantiate the increased expression of proinflammatory cytokines released by activated microglial and astroglial phenotypes in PD, while anti-inflammatory molecules experience a decline [9,10]. Consequently, a chronic and detrimental pro-inflammatory milieu is established [11]. By simulating the characteristics of human PD, preclinical studies utilizing the 6-OHDA animal model have effectively replicated these pathophysiological mechanisms, contributing to a deeper understanding of the disease processes [10].

Urgent questions are raised about the cardiovascular implications of PD with the aim of better understanding the impact of the disease on heart health and proposing strategies to improve patient outcomes. Nevertheless, the heart is still given too little attention in PD research. Our study is dedicated to examining the metabolic changes associated with disease within the heart itself. Cardiac tissue metabolomics provides a comprehensive view of the metabolic status by identifying the disease-specific metabolites and their associated metabolic pathways, thereby elucidating the biochemical mechanisms underlying the disease [12,13,14]. Here, an untargeted metabolomics approach was used to comprehensively understand the cardiac metabolic changes in the 6-OHDA rat model of PD, a widely used neurotoxic model that mimics PD-like motor and non-motor aspects, and may offer valuable insights into PD-related metabolic alterations in parkinsonian rats.

## 2. Results

### 2.1. Tyrosine Hydroxylase Immunohistochemistry

Tyrosine hydroxylase (TH) is a rate limiting enzyme in the biosynthesis of dopamine. Here, TH-positive neurons and fibers were evaluated in the SNpc and striatum, respectively (Figure 1). Significant differences were observed between 6-OHDA and sham groups. There was a 42% reduction in TH-positive striatal fibers in the 6-OHDA group compared to the sham group (Figure 2A). The 6-OHDA group also showed a 55% reduction in the TH-immunoreactive neurons in the SNpc (Figure 2B), indicating significant dopaminergic neuronal death.

### 2.2. Untargeted Metabolomics

Metabolomic analysis performed on the cardiac tissue of animals (n = 8 per group) subjected to an orthogonal partial least squares test (OPLS) in the heart revealed that there was a robust separation of metabolites between the sham and 6-OHDA groups (Figure 3A,B). The metabolites were plotted hierarchically according to their variable importance (VIP) score (Figure 3B,C) and compared to assess significant differences in their concentrations (false-discovery rate (FDR) adjusted to *p*-value < 0.05).

Levels of selected cardiac metabolites were compared to assess differences between the two groups. Lower amounts of the metabolites alanine, allothreonine and valine were identified in the 6-OHDA group compared to the sham group, while higher levels of alpha-tocopherol, beta-glycerophosphoric acid, beta-sitosterol, campesterol, stearic acid, organic phosphoethanolamine, myo- inositol -1-phosphate, monoacylglycerol and cholesterol were found in the 6-OHDA group when compared to the sham group (Figure 4). Metaboanalyst 5.0 software was used to identify the major metabolic pathways associated with the metabolites of interest (Figure 5).

## 3. Discussion

In this study, metabolomics was used to examine cardiac metabolite profiles in a rat model of Parkinson’s disease (PD) induced by 6-OHDA. Our results revealed that lipids and amino acids are key molecules implicated in abnormal metabolism related to PD. A comprehensive screening identified 12 differential metabolites between PD and sham groups. Remarkably, nine of them, namely monoacylglycerol, beta-sitosterol, campesterol, cholesterol, stearic acid, beta-glycerophosphoric acid, alpha-tocopherol, phosphoethanolamine and myo-inositol-1-phosphate, are directly associated with lipid metabolism. Noteworthy is that increased concentrations of these metabolites were observed in parkinsonian rats. The remaining three molecules belong to the category of amino acids (valine, alanine, and allothreonine) and there were consistent reductions in the levels of these metabolites in the 6-OHDA-lesioned rats.

Cardiac function relies heavily on energy metabolism, and the heart’s remarkable adaptability to switch between different metabolic substrates to fulfill its energy demands is widely recognized [15]. In terms of lipid metabolism, the heart exhibits a higher magnitude of fluxes compared to most other organs in the body, possibly rivaling the liver in this aspect [16]. Metabolomics studies in Parkinson’s disease (PD) have consistently shown significant alterations in lipid metabolism pathways, underscoring the influence of the disease on these specific metabolic processes [17]. Particularly noteworthy are the lipids with nutritional importance, including triacylglycerols (triglycerides), sterols and phospholipids. They constitute a group of fats that are essential for multiple physiological functions and serve as integral components of a healthy dietary regimen. The roles of these lipids are multifaceted, encompassing energy provision, cellular structural support, facilitation of fat-soluble molecule absorption, crucial involvement in signal transduction pathways and significant contributions to hormone synthesis, among an array of other important functions [18,19]. The pathogenesis of PD is influenced by increased inflammation and oxidative stress. Mitochondrial malfunction involves disturbed mitochondrial biogenesis and modifications in mitochondrial respiration accompanied by reduced ATP production and increased ROS generation. These pathophysiological mechanisms related to PD can potentially disrupt lipid and amino acid metabolism and impact cardiac function [20]. Therefore, metabolomics studies can reveal changes in molecules that reflect these underlying oxidative and inflammatory processes. Monoacylglycerol (MG) is involved in cellular signaling pathways and can act as a messenger in signaling cascades, particularly in processes related to inflammation, immune responses and lipid homeostasis [21,22]. On the other hand, MG can serve as an important intermediate in lipid metabolism and can be derived from the breakdown of triglycerides [23]. Increased MG levels may indicate an enhanced lipid breakdown for energy production. This can be beneficial in situations in which the organ requires additional energy, such as during certain metabolic conditions. Higher levels of MG can also influence lipid metabolism and storage processes. Excessive MG concentrations may result from imbalances in lipid metabolism, including increased synthesis or decreased breakdown of lipids. This can contribute to the accumulation of triglycerides, potentially leading to lipid-related conditions. This could potentially impact cardiac function and contribute to cardiovascular disorders. In line with our findings, a recent metabolomics study identified a significant increase in serum MG levels among PD patients when compared to a group of healthy control subjects [24].

As revealed in this present study, phytosterol levels increased in the heart of 6-OHDA rats suggesting an imbalance or dysregulation in the metabolism of these compounds. This can be influenced by various factors, including dietary intake. Phytosterols are not synthesized in mammalian cells. They are a group of naturally occurring compounds that are structurally similar to cholesterol and obtained exclusively from dietary sources, being found in nuts, seeds and vegetal oils [25]. Parkinsonian rats showed increased concentrations of both sitosterol and campesterol despite being fed with the same diet as their sham counterparts. This observation suggests that disease-related factors, such as disruptions in absorption, metabolism, excretion, tissue uptake and distribution, or biosynthesis/catabolism, may contribute to the enhanced levels of phytosterols observed specifically in the 6-OHDA-lesioned rats. Findings from animal studies on phytosterolemia have indicated cardiovascular consequences in mice, such as cardiac fibrosis, compromised cardiac function, arrhythmias and instances of sudden cardiac death [26]. Despite the recognized effect of phytosterol supplementation on lowering both LDL cholesterol and circulating triglycerides, a growing body of data suggests that higher serum concentrations of these natural plant compounds are causally associated with increased risk of cardiovascular diseases [26,27]. The significant involvement of cross-tissue/organ communication in both normal physiological processes and disease states should be noted [28]. This interorgan communication involves the transmission of signals from organs such the kidney, liver and gut to the heart, consequently impacting its function [29]. In addition to reducing cholesterol gut absorption and altering its transport, phytosterols also modulate cholesterol metabolism in the liver [30]. Maintaining the balance of cellular cholesterol levels is crucial for sustaining optimal physiological processes [31]. As a critical component of cell membranes, cholesterol impacts their fluidity and structure. It participates in intracellular transport mechanisms and serves as a precursor molecule for synthesizing vitamin D and steroid hormones [32]. However, the abnormal levels of cholesterol affect not only cardiovascular health, but also underlie an expanding array of other diseases such as neurodegenerative conditions and cancers [33,34]. Our findings indicated that PD-like pathology is associated with disturbed concentrations of cholesterol, which were observed to be elevated in the hearts of rats treated with 6-OHDA. When analyzing the impact of 6-OHDA on cholesterol concentrations, a study employing bilateral brain injection found no significant differences in plasma cholesterol levels between parkinsonian rats and control counterparts. Conversely, another study using unilateral 6-OHDA-induced lesion showed reduced concentrations of cholesterol ester in the cerebrospinal fluid of rats [35,36].

Moreover, our study identified increased amounts of both stearic acid and phosphoethanolamine in the cardiac tissue of parkinsonian rats. Of note, disease conditions can disrupt metabolic pathways, leading to changes in the synthesis, breakdown, transport of metabolites, impaired clearance or excretion, as well as enhanced metabolic demand, resulting in the accumulation of these specific metabolites in the heart. The research conducted by Shah and collaborators in the unilateral model of 6-OHDA also revealed increased levels of stearic acid in parkinsonian rats, as detected through the analysis of their plasma metabolic profile [37]. Phosphoethanolamine serves as a naturally synthesized substance that plays a crucial role in phospholipid turnover and lipid signaling pathways. Importantly, it has been shown that tissue concentrations of phosphoethanolamine are subject to a strict regulation during adulthood [38]. Kataoka and collaborators conducted an investigation to assess the developmental patterns of phosphoethanolamine expression in the heart, brain and kidney of mice at various ages ranging from 5 to 110 days. The study revealed a decline in phosphoethanolamine levels until 24 days of age, followed by a consistent concentration thereafter. The disturbed concentrations of phosphoethanolamine in the hearts of 6-OHDA-lesioned rats indicate disrupted phospholipid metabolism, suggesting altered cardiac signaling pathways in PD. A previous study that investigated the hearts of patients undergoing cardioplegic cardiac arrest revealed a substantial increase in phosphoethanolamine levels within a brief period of 20 min after cardiac arrest, suggesting a potential connection between the degradation of membrane phospholipids and myocardial pathophysiology [39].

Additionally, considering the potent antioxidant properties of α-tocopherol, the elevated cardiac concentrations of this liposoluble vitamin in 6-OHDA rats could be a potential compensatory response to counterbalance the detrimental effects induced by free radicals in the heart. Glycerol phosphates, such as β-glycerophosphoric acid (or glycerol-2-phosphate) are integral components of cell membranes along with proteins [40]. Hence, the PD-associated increase of glycerol-2-phosphate, as detected in 6-OHDA rats, has the potential to change the balance of phospholipids. This disturbance can adversely impact membrane integrity and interfere with cellular processes dependent on optimal membrane function. As a result, these alterations can impact the transport of molecules across the cell membrane, disrupt cell signaling pathways and alter the organization of membrane-associated proteins [41]. Myo-inositol-1-phosphate (MIP) contributes to the biosynthesis of inositol-containing compounds, such as inositol phospholipids [42,43,44]. Therefore, increased MIP levels, as found in PD rats, may affect the broader inositol phosphate signaling pathways. Elevated MIP concentrations may lead to an imbalance in phospholipid synthesis, particularly in the synthesis of phosphatidylinositol and its derivatives. The altered phospholipid composition can affect cell membrane structure and integrity, potentially impacting membrane-related functions such as cell signaling, transport and cell–cell interactions. Moreover, changes in MIP levels might disturb cellular osmoregulation and water balance [43].

Our study uncovered noteworthy changes in cardiac biological status related to PD. Specifically, we observed abnormal elevations in the concentrations of nine endogenous metabolites implicated in lipid metabolism. Conversely, we found reduced levels of three amino acids (valine, alanine and allothreonine) in the hearts of rats exposed to 6-OHDA, suggesting possible disturbances in amino acid uptake, synthesis and/or metabolism. These amino acids have significant roles in multiple physiological processes, encompassing energy metabolism, protein synthesis and cellular signaling [45,46]. Additionally, they are indispensable for maintaining the structural integrity of proteins, particularly those involved in contractile function and the regulation of ion channels in cardiac muscle cells. They also contribute to energy production through oxidative metabolism, providing substrates for the tricarboxylic acid cycle and ATP synthesis [47]. For instance, allothreonine fulfills pivotal roles in fat and fatty acid metabolism, in addition to its role as a substrate for the enzyme serine hydroxymethyltransferase1, which catalyzes the reversible interconversion of serine and glycine [48]. In times of energy crises, the myocardium can benefit from utilizing amino acids as alternative substrates due to their non-oxidative metabolism capabilities and lower impact on acidosis [49]. An important aspect to consider is the heart’s substantial amino acid demand, driven by the fact that the cellular proteins undergo renewal within a short timeframe [50]. Therefore, the disturbed concentrations of amino acids detected in the parkinsonian rats may have significant implications for cardiac function and overall cellular health. Reduced amino acid levels in the cardiac tissue may impact protein synthesis and turnover, impair energy production and substrate competition and affect the overall cellular machinery required for proper cardiac function [51]. They can lead to alterations in contractile performance, compromised cellular signaling pathways and potentially contribute to the development of cardiac pathologies [15,52]. A metabolomics-based study established that the assessment of plasma amino acid concentrations significantly improves the ability to predict atrial fibrillation, the prevalent type of cardiac arrhythmia [53]. Noteworthy, alterations in specific amino acids, including allothreonine and valine, were identified as valuable diagnostic indicators of atrial fibrillation [54].

The MetaboAnalyst platform was used to identify the relevant metabolic pathways to interpret our results in a biologically meaningful scenario from the concentration patterns of cardiac metabolites that discriminated sham and parkinsonian rats. Several metabolic pathways, including phosphatidylethanolamine biosynthesis, the glucose-alanine cycle, phosphatidylcholine biosynthesis, alanine metabolism, glutathione metabolism, plasmalogen synthesis and inositol phosphate metabolism were identified from the analysis.

The biosynthesis of both phosphatidylethanolamine (PE) and phosphatidylcholine (PC) has been identified as relevant metabolic pathways in the present study. Crucial for a great variety of cellular processes, the glycerophospholipids PC and PE are the two most abundant phospholipids in eukaryotic cells, and, in addition to their central roles in the structure and function of membranes, they are essential regulators of lipids and cell energy metabolism [54,55]. The cytidine diphosphate (CDP)- ethanolamine and CDP-choline are branches of the Kennedy pathways employed by mammalian cells for PE and PC biosynthesis in the endoplasmic reticulum, representing major biosynthetic routes in the formation of these phospholipids [56,57]. Alterations in the tissue expression of PC and PE have been implicated in metabolic disorders including atherosclerosis [58]. Comprising roughly 20% of membrane phospholipids, plasmalogens are a particular subclass of glycerophospholipids containing a sn-1 vinyl ether linkage plus an ester bond at the sn-1 and sn-2 positions of the glycerol backbone, playing a diversity of cellular functions [58]. Like other lipid mediators, plasmalogens are vital molecular messengers involved in inflammatory responses and defects in their synthesis are implicated in metabolic disorders and neurodegenerative diseases [59,60].

The metabolism of phospholipids is closely linked with that of other lipids within the cell. The glucose–alanine cycle, otherwise known as the alanine (a glucogenic amino acid) cycle or Cahill cycle, links carbohydrate metabolism to amino acid metabolism and mediates the degradation of muscle protein in order to deliver extra glucose to produce more ATP to sustain muscle contraction, regulating energy metabolism [61,62]. Found in all mammalian tissues, glutathione is a rich antioxidant synthesized through two sequential reactions in the cytosol through the action of ATP-dependent enzymes: first, γ-glutamylcysteine is formed from L-glutamate and cysteine, catalyzed by glutamate–cysteine ligase; the second reaction produces glutathione by adding glycine to the C-terminal of γ-glutamylcysteine, a step catalyzed by glutathione synthase [63]. Glutathione is involved in biological processes, such as redox reactions and cell homeostasis, and is a modulator of cell proliferation, immune response, apoptosis and others [64]. Furthermore, glutathione system dysfunction has been implicated in brain disorders, including PD [6,65]. Eukaryotic organisms have developed intricate signaling networks to effectively respond and adapt to the dynamic nature of their environment. A notable system in this context is the group of soluble and membrane-associated inositol phosphates that coordinates diverse biological activities, spanning from nutrient uptake and utilization to growth factor signaling and the maintenance of energy homeostasis [66,67]. The inositol phosphates are multifunctional molecules represented by a family of mono- to poly-phosphorylated inositols, including the monophosphorylated inositol to the inositolhexakisphosphate [68]. The biological and pharmacological properties of inositol phosphates rely on the specific phosphate group content in the inositol ring, which is required for the precise propagation of cellular information. Moreover, the phosphorylated derivates of inositol phosphates are inositol pyrophosphates, a class of high-energy molecules with a range of metabolic and signaling functions [66,69].

Extensive research is still being conducted today to unravel the enigma of PD, aiming to acquire a more comprehensive understanding of the condition, devise novel therapeutic approaches and optimize patient outcomes. The diagnosis of PD puts patients at an elevated risk of developing cardiovascular diseases. The heart and its functions are profoundly compromised in PD. Nonetheless, there is a lack of comprehensive knowledge about the specific impact of PD on the heart itself. Our study found that lipids (phospholipids), amino acids and energy metabolism are deeply implicated in cardiac metabolic changes related to PD. These results provide new insights into cardiac metabolism in experimental PD and indicate potential targets for further investigation.

As complex multicellular organisms, mammalian metabolism involves interconnected organ systems. In this study, we focused on the metabolic changes associated with PD in a specific tissue, the heart. Future studies should adopt a more integrated approach focused on exploring inter-tissue communication. Therefore, the next steps to gain new insights into PD pathogenesis should involve applying a network-based approach to uncover PD related multi-tissue crosstalk. Future advancements in omics and multi-omics research complemented by the systematic study of biological species, including samples from PD patients, will advance our understanding and lead to a greater recognition of the importance of the regulatory components for this disabling disease.

## 4. Materials and Methods

### 4.1. Animals

The rats were provided by the Center for Development of Experimental Models for Medicine and Biology (CEDEME)/Universidade Federal de São Paulo and were kept in the bioterium of the Neuroscience laboratory, housed in groups of 4 per cage with sawdust as bedding. The cages were maintained under a 12 h light-dark cycle (light: 7:00–19:00), temperature at 21 ± 2 °C with food and water ad libitum. Sixteen adult male Wistar rats weighing between 230 and 300 g and 8 weeks of age were used. Animals were distributed in two experimental groups: sham and 6-OHDA. All experiments followed the guidelines of the Ethics Committee of the Federal University of São Paulo to avoid unnecessary animal suffering. Routinely, the vivarium is visited by the responsible veterinarian, who assesses the health of the animals.

### 4.2. Study Design

The rats were randomly divided into two groups. In the 6-OHDA group, the animals received bilateral injections of 6-OHDA in the striatum, while the sham group received saline solution (vehicle). After 40 days, the brains and hearts were removed for immunohistochemistry and metabolomics, respectively.

### 4.3. 6-OHDA Lesion

Fifteen minutes before stereotactic surgery, the animals were anesthetized intraperitoneally with a single injection of ketamine (100 mg/kg) and xylazine (10 mg/kg) and after complete anesthesia, they were fixed in a stereotaxic device (EFF 331-Insight™, RibeirãoPreto, São Paulo, Brazil). A 10 µL Hamilton syringe was attached to the stereotaxic rod to inject 1 µL of 6-OHDA solution (Sigma©, Saint Louis, MO, USA) (vehicle: 0.3% ascorbic acid). Rats received two injections of 6 µg/µL of 6-OHDA per striatum, totalizing 12 µg in each brain hemisphere, or received saline solution. Four different injection coordinates relative to the bregma were obatined from Paxinos and Watson’s brain atlas [70] as follows: (1) latero-lateral: −2.7 mm, anteroposterior: bregma, dorsoventral: −4.5 mm; (2) latero-lateral: −3.2 mm, anteroposterior: +0.5 mm, dorsoventral: −4.5 mm; (3) latero-lateral: +2.7 mm, anteroposterior: bregma, dorsoventral: −4.5 mm; and (4) latero-lateral: +3.2 mm, anteroposterior: +0.5 mm, dorsoventral: −4.5 mm. Bilateral administration of the neurotoxin 6-OHDA, which is structurally similar to catecholamines, to the dorsal striatum causes death of dopaminergic neurons in the substantia nigra pars compacta (SNc), mimicking Parkinson’s pathology [71,72].

### 4.4. Immunohistochemistry

The rats were anesthetized with lidocaine (10 mg/kg i.p.) and sodium thiopental (80 mg/kg i.p.). After complete anesthesia, the rats were decapitated, their hearts were removed and quickly frozen with liquid nitrogen and then stored in a freezer at −80 °C until use. The rat brains were removed and placed in 4% paraformaldehyde solution for 24 h for fixation and then transferred to phosphate-buffered saline (PBS) (0.01 M pH 7.4). Finally, the brains were immersed in a 30% sucrose cryoprotective solution for 2 days, allowing a 40 μΜ coronal cut with a cryostat (Microm HM 505E, Ramsey, NJ, USA). Brain tissue slices were washed for 5 min in 0.01 M PBS (pH 7.4), treated with 0.1% hydrogen peroxide to remove endogenous peroxides and, after washing, incubated in 10% albumin solution and 0.3% Triton X-100 for 2 h. After this period, the slices were incubated overnight with primary antibody (1:500 tyrosine hydroxylase (TH), diluted in 0.01 M PBS (pH 7.4) and 2% albumin. The brain slices were treated with biotinylated secondary antibody (anti-rabbit 1:200-Abcam^®^, Waltham, MA, USA) diluted in 0.01 M PBS and 2% albumin per 2 h. The tissue was then washed and incubated with Avidin-Biotin-Peroxidase complex (ABC Elite; Vector Labs, Burlingame, CA, USA). The sections were stained with 3,3’-Diaminobenzidine tetrachloride (DAB) dissolved in TRIS-HCl 0.05 M (PH 7.6) and activated by 0.3% hydrogen peroxide. The images were analyzed using the free ImageJ program to indicate cell nuclei and fibers in the striatum and substantia nigra region.

### 4.5. Metabolomics

For the metabolomic analyses, the hearts stored in the ultra-low freezer (−80 °C) were removed and homogenized in a suitable solution for disrupting lipid molecules (TissueLyser; Qiagen, Germantown, MD, USA). For gas chromatography analysis (GC-MS), aliquots of 100 µL samples from both groups were shaken with pure methanol in a 1:3 ratio in such a way that deproteinization occurred and centrifuged at high speed, and 100 µL of the supernatant was placed in GC-MS vials with glass pellets for the derivatization process. In the methoximation step, the solvent was evaporated at 30 °C in a SpeedVac and the O-methoxyamine hydrochloride (15 mg/mL) in pyridine was placed in the flasks, vortexed and incubated for 16 h (dark and at room temperature). After this time, silylation with 10 µL of BSTFA (1% TMCS (*v/v*)) started and the samples were incubated at 70 °C for 1 h. Finally, 100 µL of heptane (with 20 ppm of internal standard pentadecanoic acid) was added to each analysis vial. Blanks were processed together to correct the baseline chromatograms [73]. Analyses were performed on a quadrupole-type GCMS-QP2020NX system (Shimadzu Co., Kyoto, Japan), with 1 uL of the sample loaded onto a DB5-MS column (30 m × 0.25 mm, 0.25 µm, Restek, Bellefonte, PA, USA) and injected in splitless mode with helium gas flow (20 mL/min). Carrier gas was conducted at a constant flow of 1.36 mL/min. The initial column temperature was 100 °C followed by increasing the temperature at 15 °C/min until it reached 300°C. The temperature was then left at 300 °C for five minutes before cooling. The injector (280 °C), transfer line (200 °C) and quadrupole source filament (150 °C) temperatures remained unchanged. The system operated in full scan mode (40–650 m/z and 3 spectra/s) with the EI set to 70 eV. LabSolutions software (GCMS version 4.5, Shimadzu Co., Kyoto, Japan) allows real-time control of each analyte analyzed to label the metabolites in the Scan. The detected metabolites were processed to create a unified matrix with the different charge states, adducts and groups of the same analytes from all samples using GCMS Solution (v.3.30), NIST 17 MASS (v.1.00.1) and GCMS software Smart Metabolite (v.3.01), all developed by Shimadzu Co. After identification of molecules by NIST 14 and Smart Metabolite libraries, samples were exported to Excel (v16.0 Microsoft Office© 2023, Redmond, WA, USA) software for statistical treatment using the free online tool available metaboanalyst 5.0. Public databases available on the internet, such as the Human Metabolome Database (https://hmdb.ca/) were used (https://www.genome.jp/kegg/, accessed on 15 December 2022

### 4.6. Statistics

Since the Shapiro–Wilk test showed that the immunohistochemistry was not normally distributed, the study groups were compared simultaneously with the non-parametric Kruskal–Wallis test. Bonferroni’s post hoc test determined whether the groups were significantly different from each other. The metabolomic data were normalized, and the parametric ANOVA test was used to simultaneously compare the two groups. Tukey’s post hoc test determined whether there were significant differences between the groups. The level of statistical significance was set at 0.05. Statistical tests were performed using MATLAB R2017a, GraphPad Prism 5.0 and Metaboanalyst 5.0.

## Figures and Tables

**Figure 1 ijms-24-12202-f001:**
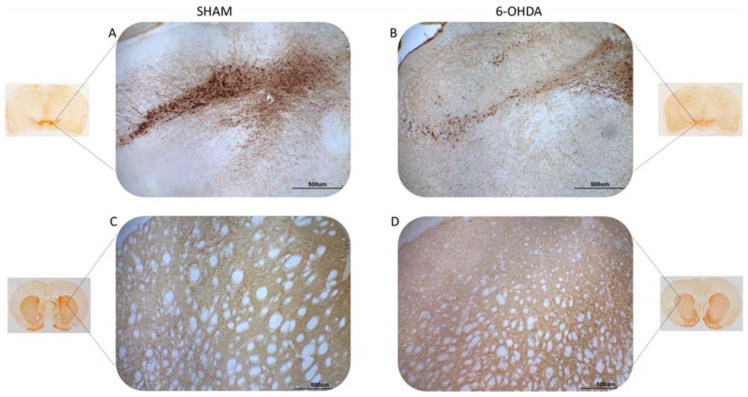
Illustrative images of tyrosine hydroxylase (TH) immunoreactivity. TH-positive neurons (**A**,**B**) and fibers (**C**,**D**) in the SNpc and striatum, respectively. Sham group (**A**,**C**) and 6-OHDA (**B**,**D**). Note the evident reduction of TH-immunostaining in brain slices from the 6-OHDA group compared to the sham group. The scale line in each image is equivalent to 500 micrometers.

**Figure 2 ijms-24-12202-f002:**
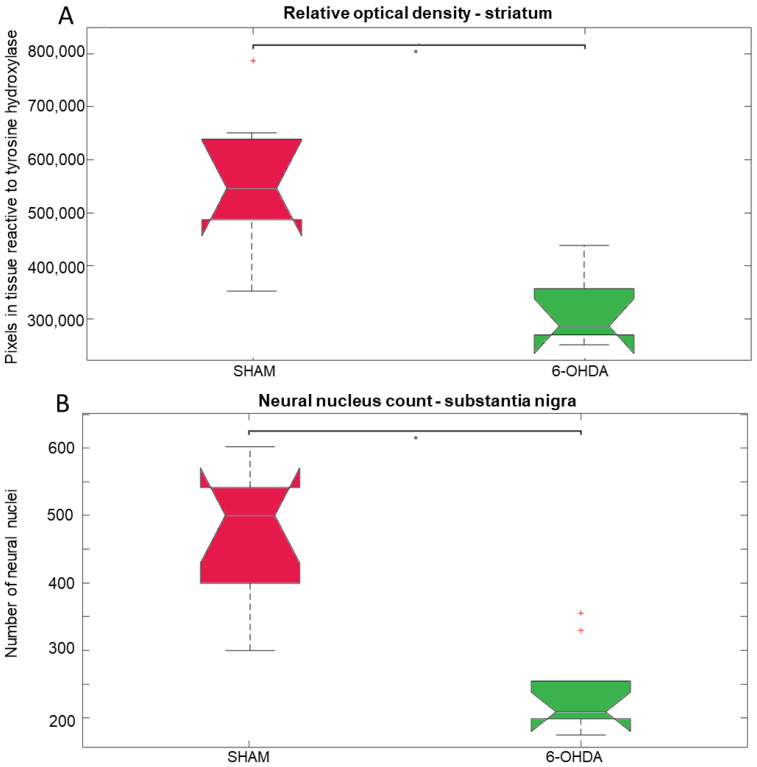
Statistical results of the Kruskall–Wallis and post hoc Bonferroni for tyrosine hydroxylase (TH) immunohistochemistry analysis. (**A**) Optical density of the TH-positive fibers in the striatum (chi-square = 8.27, *p* = 0.004); median and confidence interval were as follows: sham (550.040 ± 86.764) and 6-OHDA (299.536 ± 49.874). (**B**) TH-positive neuronal nuclei in the substantianigra pars compacta (chi-square = 11.76, *p* = 0.0006); median and confidence interval were as follows: sham (501 ± 71) and 6-OHDA (212 ± 29). The * represents significant difference with *p*-value < 0.05. Red crosses indicate values outside the confidence interval for the median.

**Figure 3 ijms-24-12202-f003:**
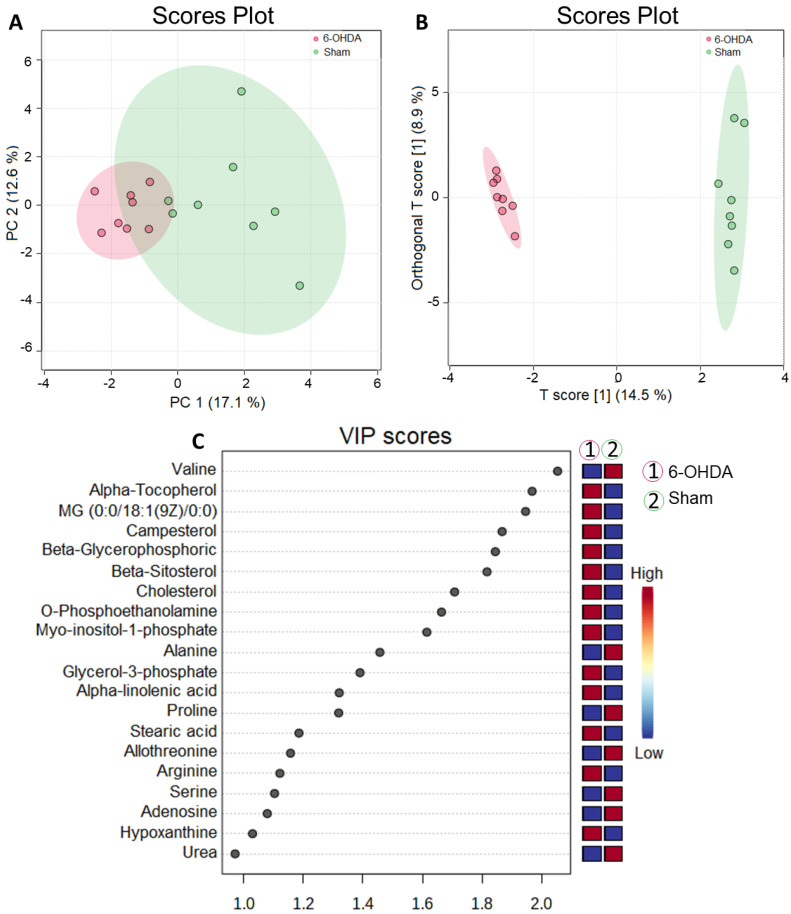
Multivariate statistical analysis of metabolites between 6-OHDA and sham groups. (**A**) PCA analysis to identify the most important variables (PC1, main component one; PC2, main component two) between the two groups and ensure data reliability. (**B**) PLS plot showing the separation of groups according to their different characteristics in relation to the metabolites. (**C**) Main metabolites identified in the cardiac tissue of animals, presented in VIP score.

**Figure 4 ijms-24-12202-f004:**
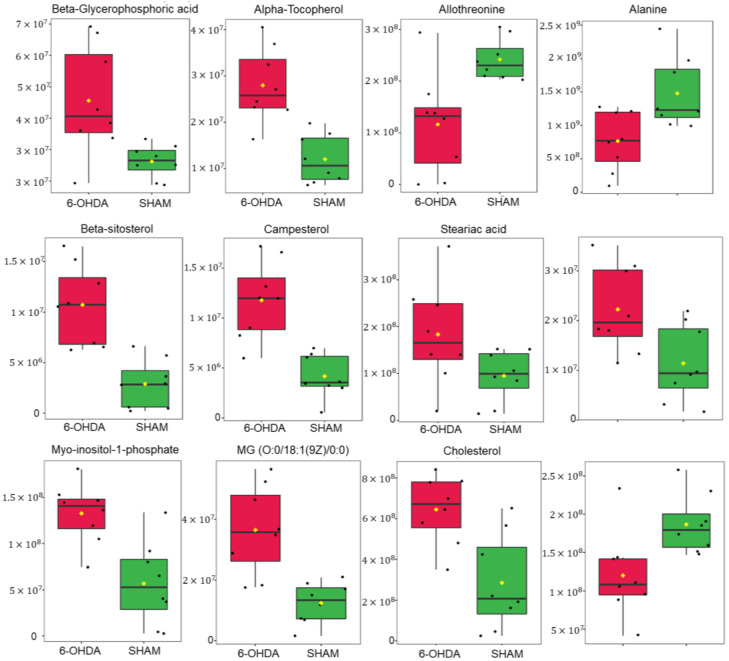
Statistical analysis of metabolites with higher amounts in the comparison between the two groups. The (*p*-value, FDR) for the metabolites are: beta-glycerophosphoric acid (0.04662, 0.035601); alpha-tocopherol (0.001088, 0.020886); allothreonine (0.004662, 0.035601); alanine (0.006993, 0.035601); beta-sitosterol (0.01865, 0.020886); campesterol (0.001865, 0.020886); stearic acid (0.010412, 0.048588); O-phosphoethanolamine (0.006993, 0.035601); myo-inositol-1-phosphate (0.006993, 0.035601); MG (0:0/18:1(9Z)/0:0) (0.001088, 0.020886); cholesterol (0.006993, 0.035601); valine (0.001865, 0.020886). The data distribution is represented by the black dots, with the mean indicated by the yellow dot.

**Figure 5 ijms-24-12202-f005:**
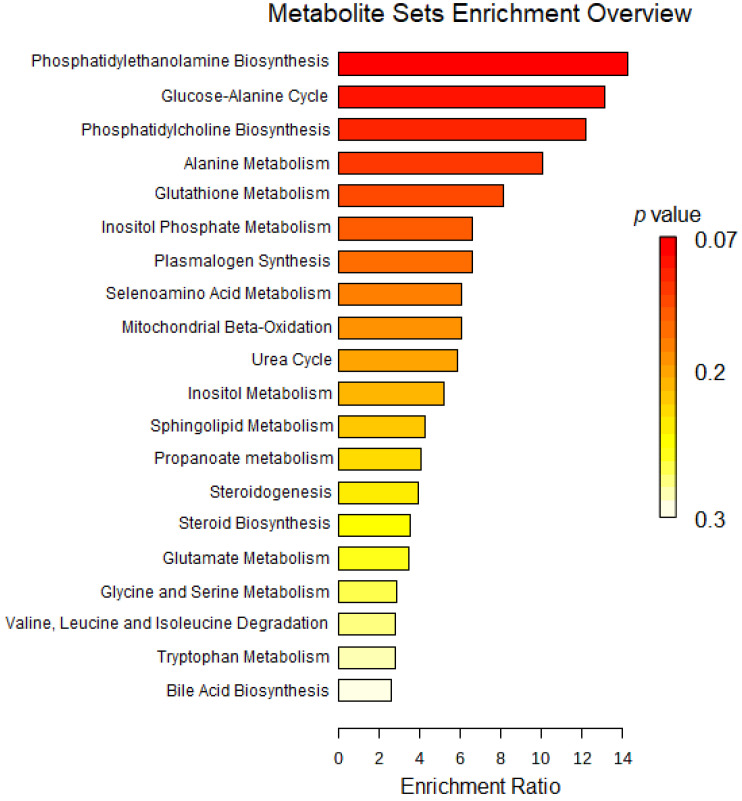
Analysis of metabolite enrichment between 6-OHDA and sham groups. The relevant metabolic pathways to the interpretation of the concentration patterns of the metabolites of interest are illustrated.

## Data Availability

Not applicable.

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
