# Peer review of "Parkinson’s Disease and the Heart: Studying Cardiac Metabolism in the 6-Hydroxydopamine Model"

_ijms, 2023, doi:10.3390/ijms241512202_

Round 1
Reviewer 1 Report
The manuscript entitled "Parkinson's Disease and the Heart: Studying Cardiac Metabolism in the 6-Hydroxydopamine model" describes the metabolomic analysis of a rat model of PD. The PD phenotype was induced by bilateral injection of 6-hydroxydopamine (6-OHDA) in the striatum of the model animals. From the description, it is not entirely clear if the total amount of 6-OHDA injected was 12 ug, or if 4 x 12 ug were injected(in 4 different coordinates). Control (sham) animals were injected with saline solution.
40 days after injection, the animals were sacrificed and the brains and hearts analyzed. Successful induction of the PD phenotype was assessed by immunohistochemical analysis of the brain, probing for tyrosine hydroxylase-positive neurons and fibers. Significant reduction of immunoreactive striatal fibers and neurons in the 6-OHDA treated rats was the evidence indicating the desired PD-like outcome.
Differences of metabolite content of hearts of 6-OHDA - treated animals and controls was determined by gas chromatography - mass spectrometry analysis.
Abundance of several metabolites showed clear differences between hearts of treated vs sham animals. In particular, phospholipids, amino acid - and energy metabolism appears to be affected. The results are clear and convincing.
However, the very elaborate discussion reads more like a review of literature on the metabolites discussed than an attempt to rationalize the reasons for the observed changes. I cannot recognize any attempt to formulate a mechanism that would explain the observed changes. While the article is OK as a list of observed change and the analyses appear to be done expertly, it is my opinion that this is not sufficient for a publication in IJMS.
The manuscript is written in good English, there are only a few places I found that need editing. Specific remarks:
Lines 54 -55: "..and provide valuable insights into disease-associated with metabolic changes in Parkinsonian animals." There is something wrong with this sentence
Line 162: "synthetized" should be "synthesized" or "synthesised" depending if you prefer the UK or US spelling
Legend to Figure 4: it says "The * represents significant difference with p-value < 0.05." (line 120) - I cannot see the star referred to anywhere in the figure
Author Response
Good afternoon,
Attached is the file with acknowledgments and revisions.

Reviewer 2 Report
The article entitled Parkinson's Disease and the Heart: Studying Cardiac Metabolism in the 6-Hydroxydopamine Model is an interesting article in which the authors used metabolic pathway analysis to determine whether different metabolites and studied their involved pathway-taking Parkinsonian rat model. The author reached the conclusion that both lipid and energy metabolism are associated with cardiac metabolism in PD suggesting pinpointing the need to advance this field of research.
However, there is some weakness in the article such as
1. Introduction section: Line 44- Line 48
The introduction section is very short. I suggest expanding your introduction section by adding another 1-2 paragraphs. The context that you may introduce here can be introduced from the below suggestions.
a. Explain inflammatory molecules or cascade in Parkinson's disease (Cytokine, microglia/astrocytes context etc).
b. Explain the antioxidant system such as glutathione and its involvement in Parkinson's disease.
C. Introduce glia context in Parkinson's disease.
This will give a more generalized and wide concept of the field.
d. Line 48: Please cite a few more references that show a link between cardiac tissue metabolomics that have an influence on the neuronal signaling cascade in the brain.
2. Figure 2 or elsewhere: If applicable please provide the respective unit (Y-axis). In particular, when there is a value in the y-axis in thousands, may be better to quantify in log value or exponential power form.
It may be better to place "*" or another symbol above the line that shows the p-value significant.
3. In Figure 3 (left picture), lines 91-92: The text in the figure is too crowded. I suggest increasing the font size and figure symbol with bigger font. Please specify those symbols' names in the index or legends.
4. Figure 4 or all of the figures: Wherever applicable, you should provide the readable text in the figures (including the y-axis, x-axis, or wherever applicable).
5. Line 307: As suggested earlier in the introduction section from the literature, please expand how the multi-tissue or omnidirectional cross-talk perspective here.
6. Line 313-321: Please specify if the veterinarian is available to consult if the animal gets sick or looks at any issues.
7. Line 360: Check if "avidinbiotin-peroxidase" is correctly written.
Author Response

(The authors gave the same response as above.)

Round 2
Reviewer 1 Report
Overall, this is much improved, with background, motivations and conclusions better worked out. The discussion looks less like a laundry list now. I would still wish they could provide a mechanism of linkage of what is primarily a neurodegenerative disorder to heart functions and changes in heart metabolism. Alas, this might be a bigger project than can be tackled in this one publication.
The language is good;
One type to fix: lines 162-163: infuence --> influence
Author Response
We would like to thank you immensely for all the considerations. All authors agree with the improvement of the article. Thanks also for pointing out the spelling error, it has been corrected.
Reviewer 2 Report
The article has slightly improved now, but it still created some imbalances or weaknesses. Please fix it before it goes for publication.
1. If you have difficulty in constructing readable documents (Figure 3C), this figure should be omitted. I agree that you must remove Figure 3 c.
2. All other figures and symbols used in the figure should be presented in bigger font size and readable without difficulty from a distance as I mentioned earlier in my comments. If you compare the text in the x-axis and y-axis or text around the figure you would realize such inconsistency. Some text looks hazy (not focused). You MUST fix those text making it more clear figures.
3. There is no balance in the citations now with the addition of updates, this article has hugely cited all the articles arising from MDPI publishers (such as IJMS, Cells, etc), please provide a balanced citation arising from diverse journals articles from the most recent years.
Author Response
Dear reviewer, we greatly appreciate all corrections and suggestions made. Below are our changes and considerations.
About the figures:
Figure 1: there is a line and a number inside each image, it is a scale line generated by the microscope program, according to the lens used. We've added a sentence to the caption to make it easier to understand, in case it's hard to see the accompanying number: "The scale line in each image is equivalent to 500 micromers."
Figure 3: figure 3c (presented in the previous version) was removed, due to difficulty in reading. The other figures have been rearranged and the axes enlarged.
Figure 3 and 4: legends and axes have had their fonts enlarged.
Finally, new references were added to the text and are indicated in yellow for better location.